# Study on the Microstructure and Magnetic Properties of Nd-Fe-B/Fe-Co Composite Nanowires

**DOI:** 10.3390/ma16165541

**Published:** 2023-08-09

**Authors:** Licong Kang, Wei Yang, Lichen Zhao, Chunxiang Cui, Feng Cao

**Affiliations:** 1Hebei Key Laboratory of New Functional Materials, School of Material Science and Engineering, Hebei University of Technology, No. 5340, Xiping Road, Tianjin 300401, China; klc910808@126.com (L.K.); zhlch@hebut.edu.cn (L.Z.); hutcui@hebut.edu.cn (C.C.); 2State Nuclear Power Demonstration Plant Company Limited, No. 666, Fujia Road, Weihai 264200, China; caofeng@spic.com.cn

**Keywords:** Nd-Fe-B, Fe-Co, electrochemical deposition, magnetic properties, exchange coupling

## Abstract

To solve the problem of the low coercivity of Nd-Fe-B-based nanowires impeding their application in magnetic storage media, highly ordered Nd-Fe-B/Fe-Co composite nanowires were fabricated in an anodic alumina template by means of the alternating electrochemical deposition method. In this paper, the effect of soft and hard magnetic phase compositing on the magnetic properties of Nd-Fe-B-based nanowires was investigated, and the coercivity improvement mechanism was demonstrated. The results show that after annealing at 600 °C for 2 h, Nd-Fe-B/Fe-Co nanowires crystallize into a multiphase structure containing a hard Nd_2_(Fe, Co)_14_B phase and soft NdB_4_, NdB_6_, Fe_7_Nd, and Fe_7_Co_3_ phases. It is characterized that the Nd_2_(Fe, Co)_14_B phase preferentially nucleates, followed by NdB_4_ + NdB_6_ + Fe_7_Nd, while Fe_7_Co_3_ has been formed in as-deposited nanowires. The existence of a Nd_2_(Fe, Co)_14_B phase with high anisotropy fields, the remanence enhancement effect produced by exchange coupling between hard–soft magnetic phases, and the pinning effect between different phases make the composite nanowires approximately exhibit single hard magnetic phase characteristics with coercivity and remanence ratio as high as 4203.25 Oe and 0.89. The results indicate that synthesizing Nd-Fe-B/Fe-Co exchange-coupled composite nanowires via alternating electrodeposition is an effective way to optimize the magnetic performance of Nd-Fe-B-based nanowires.

## 1. Introduction

Magnetic nanowires with large aspect ratios and apparent shape anisotropy are one of the research focuses in the new magnetic functional materials field, which has good application prospects in high-density linear magnetic storage media. As the third-generation rare earth magnets, Nd-Fe-B has outstanding magnetic properties that other materials cannot surpass. However, the low coercivity of currently manufactured Nd-Fe-B nanowires hinders their application in magnetic storage [1,2,3]. Therefore, how to prepare Nd-Fe-B-based nanowires with high magnetism has certain theoretical significance for developing new-generation vertical magnetic storage media. There has been a great deal of research on how to improve the hard magnetism of Nd-Fe-B nanowires through doping alloy elements [4,5,6,7,8,9,10,11]. In order to be better suitable for high-density linear perpendicular magnetic recording media, in addition to improving the coercivity of Nd-Fe-B nanowires, higher saturation magnetization should be expected to enhance their storage density.

The fabrication of nanocrystalline composites is an effective method to obtain both high coercivity and high saturation magnetization, which could obtain high magnetic energy products through the exchange coupling of hard–soft magnetic phases [12,13,14,15]. Fe-Co alloy is an excellent soft magnetic material that has a high saturation magnetic moment and initial permeability [16,17,18]. At the same time, the low coercivity and remanence ratio of soft magnetic materials limits their application in magnetic storage media [19]. Mansouri et al. [20] fabricated FeCo and FeNi nanowires with a coercivity of 627 Oe and 279 Oe through electrochemical deposition. Han et al. [21] successfully prepared a series of Fe_x_Co_1−x_ (x = 0.0, 0.5, 1.0) nanowires with coercivity in the range of 450 Oe~570 Oe and a squareness of 0.23~0.41. Kaniukov et al. [22] researched the structural and magnetic performances of FeCo-alloy nanotubes deposited under different potentials and found that the nanotubes exhibited more excellent properties with a coercivity of 500 Oe and squareness of 0.2 due to the high crystallinity degree under the higher potential of 2 V. It is observed that most Fe-Co magnetic nanowires exhibit typical soft magnetic properties with low coercivity and a low magnetic energy product.

To simultaneously possess high coercivity and high saturation magnetization, there have been many studies about the combination of Fe-Co and rare earth permanent magnetic alloys. Chuanto et al. [23] fabricated SmCo_5_/FeCo core–shell composite materials, showing how the magnetic energy product of a composite material with 10 wt.% FeCo increased by 25% compared to a single hard magnetic phase SmCo_5_. Cui et al. [24] prepared Sm_2_Co_17_/Fe_7_Co_3_ duplex-phase magnetic nanowires with excellent shape anisotropy via alternating electrodeposition. They found that the duplex-phase Sm_2_Co_17_/Fe_7_Co_3_ magnetic nanowires have better remanence and coercivity in both annealed and deposited states than Fe_7_Co_3_ and Sm_2_Co_17_ nanowires. However, research on Nd-Fe-B-based nanocomposites mainly focuses on composites with soft magnetic α-Fe and Fe_3_B phases, currently [25,26,27]. Therefore, Nd-Fe-B/Fe-Co composite nanowires are an exciting route to solve the low coercivity of Nd-Fe-B-based nanowires.

The present work demonstrates an alternating electrochemical deposition for fabricating aligned Nd-Fe-B/Fe-Co composite nanowires within AAO templates. It is expected that a Nd-Fe-B/Fe-Co hybrid magnet, which is more suitable for magnetic storage media, can be obtained through soft–hard magnetic exchange coupling. This article also discusses the effect of the Fe-Co alloy on the phase component, microstructure, and properties of the Nd-Fe-B/Fe-Co composite nanowires before and after annealing. Moreover, this article presents the evolution of microstructure phase transition and phase distribution. And finally, it investigates the effects of phase evolution, dislocation, and the coupling of the hard–soft phase exchange on the magnetic performance of Nd-Fe-B/Fe-Co nanowires.

## 2. Materials and Methods

The methodology flow chart of this work is illustrated in Figure 1. The detailed procedure is briefly described below. 

### 2.1. Synthesizing Process of AAO Template

The highly ordered alumina (AAO) templates used in this experiment were prepared via secondary anodizing in 0.3 mol/L oxalic acid solution under 40 V DC. During the anodization process, the oxidation tanks were placed in an ice–water mixture, preventing the excessive temperature caused by the exothermic reaction from affecting the quality of the oxide film. Between the first (4 h) and second (6 h) anodization, we used chromic acid and phosphoric acid to remove the oxide layer with lower order generated during the first anodization process. Highly ordered AAO templates with a pore size of 70 nm and length of 30 μm were successfully prepared through removing the bottom and reaming (Figure 2). It can be seen that AAO templates contain self-assembled, highly uniform quasi-hexagonal ordered holes. As shown in Figure 2c,d, the pore channels are parallel to each other and perpendicular to the membrane surface.

### 2.2. Electrochemical Deposition of Nd-Fe-B/Fe-Co Composite Nanowires

Before electrodeposition, a thin layer of gold film was plated on the back of the template to improve the conductivity of the alumina template. The deposition was based on graphite as the anode and the AAO template as the cathode, conducted under 2 V DC. The alternating deposition was completed through depositing in Fe-Co electrolyte for 1 min followed by Nd-Fe-B electrolyte for 30 min, alternating deposition three times. Table 1 lists the composition of the Fe-Co and Nd-Fe-B solutions. The deposition process was carried out on a magnetic stirrer, and the deposited solution was constantly stirred to accelerate ion diffusion and ensure a uniform concentration of ions in the electrolyte. A portion of the nanowires within the AAO template was then annealed at 600 °C for 2 h in a vacuum heat treatment furnace at a heating rate of 5 °C/min in an atmosphere of 99.99% argon.

### 2.3. Characterization

The phase composition of the samples was analyzed using a Bruker D8 Discover X-ray diffractometer (XRD) (Bruker, Billerica, MA, USA) with Cu kα radiation (λ = 1.5406 Å), a voltage of 40 kV, and a current of 20 mA. In the range of 2θ from 20° to 90°, the XRD patterns were scanned with a step length of 0.02°and a scanning rate of 6°/min. The chemical compositions and morphologies of nanowires with partially dissolved AAO templates were characterized using Quantum 450 FEG scanning electron microscopy (Beijing, China) (SEM) and energy dispersive spectroscopy (EDS). The microstructure of the nanowires was observed using the Tecnai G2 F30 S-TWIN transmission electron microscope (TEM) (Thermo Fisher Scientific, Waltham, MA, USA). The magnetic properties of nanowires in the AAO templates were measured using a Lake Shore 7407 type vibrating sample magnetometer (VSM) in a maximum 15 kOe magnetic field parallel to the long axis of the nanowires.

## 3. Results

### 3.1. Phase Analysis

Figure 3a,b show the XRD patterns of Fe-Co, Nd-Fe-B, and Nd-Fe-B/Fe-Co composite nanowire arrays before and after annealing. It can be seen from Figure 3a that the as-deposited Nd-Fe-B/Fe-Co nanowires consist of a Fe_7_Co_3_ phase. Compared with Fe-Co nanowires, there is a higher amount of amorphous diffuse scattering peaks in the XRD patterns of Nd-Fe-B and Nd-Fe-B/Fe-Co nanowires, which is due to the existence of a Nd atom with a large radius improving the amorphous forming ability of the nanowires. 

Figure 3b shows the XRD patterns of nanowires annealed at 600 °C for 2 h in high-purity Ar. In comparison, the diffraction peak intensity for annealed nanowires increases and half-width decreases, indicating that the grains’ size and crystallinity improved after annealing [28]. The 600 °C-annealed Nd-Fe-B/Fe-Co nanowires present a complex phase composition containing the newly formed Nd_2_(Fe, Co)_14_B, NdB_4_, NdB_6_, and Fe_7_Nd phases as well as the Fe_7_Co_3_ phase existing in as-deposited nanowires. The formation of the Nd_2_(Fe, Co)_14_B phase illustrates that in addition to forming the Fe_7_Co_3_ phase with Fe atoms during the alternating electrodeposition process, some Co atoms also enter the Nd_2_Fe_14_B main phase through replacing some Fe atoms during heat treatment. As a result of the substitution of Co (0.126 nm) with a smaller atomic radius for Fe (0.127 nm), lattice contraction occurs in the 2:14:1-type phase according to the Bragg equation, λ=2dsinθ, leading to the right shift of the 2:14:1-type phase diffraction peak as seen from the example of 204 peaks in the inset of Figure 3b, in which the right shift is approximately 0.4°.

Compared with Nd-Fe-B nanowires, there are more diffraction peaks of the 2:14:1 type phase in the XRD pattern of Nd-Fe-B/Fe-Co nanowires, which is related to the deposition efficiency of Nd atoms and their structural evolution. (1) Rare earth metals are difficult to reduce due to their low potential, which can be realized through the co-deposition of an RE-TM alloy. The transition metal Fe and Co ions in Fe-Co solutions with similar structural properties and positive electrode potential are easily reduced and deposited into the AAO template. The generation of a large number of Fe and Co atoms promotes dragging more Nd ions into AAO template pores during the alternating electrodeposition process, inducing more Nd atoms to realize co-deposition, thereby improving the deposition efficiency and quantity of Nd atoms.

Assuming that the phase transition process only considers the energy required for phase crystallization without other heat, the phase Gibbs free energy ΔG (i.e., the latent heat of phase transition from amorphous to crystalline) can be considered to be approximately equal to the phase formation enthalpy ΔH. During multiphase structure transformation, it is easier for the phase with smaller ΔG to overcome the energy barrier, then nucleate and grow up, completing phase transition. Based on this, the thermodynamic formula of the Arrhenius Equation can be extended and obtained from the following formulas [29]:(1)$$\ln k_{2i}/k_{1i}=-\Delta H_{i}\left(1/T_{2}-1/T_{1}\right)/R\;(i=1,\;2,\;3,\;4,\;5)$$
(2)$$k_{1i}=I_{\max i}{}_{\left(843\;K\right)}/t,\;k_{2i}=I_{\max i}{}_{\left(873\;K\right)}/t$$
(3)ΔG=ΔH
wherein $k_{1i}$ and $k_{2i}$ are the crystallization rates for the phase in nanowires annealed at different temperatures of *T*_1_ (570 °C) and *T*_2_ (600 °C), respectively. *I*_max*i*_ is the maximum diffraction peak intensity of each phase in annealed nanowires. The subscript *i* represents the phase formed during the annealing process, and *R* is the gas constant (8.314 J/(mol·K)). 

Figure 4a,b show the XRD patterns of Nd-Fe-B/Fe-Co and Nd-Fe-B nanowires annealed at temperatures of T_1_ (570 °C) and T_2_ (600 °C), respectively. Table 2 lists the *I*_max_ of each phase in nanowires. The ∆G of Nd_2_(Fe, Co)_14_B, NdB_4_, NdB_6_, Fe_7_Nd, and Fe_7_Co_3_ phases in Nd-Fe-B/Fe-Co composite nanowires are calculated using Formulas (1)–(3) to be 29,188.6 J/mol, 90,461.4 J/mol, 84,392.5 J/mol, 82,220.4 J/mol, and −46,742.8 J/mol, respectively. The ∆G (Fe_7_Co_3_) < 0 indicates that the Fe_7_Co_3_ phase can spontaneously nucleate and crystallize, consistent with the XRD results shown in Figure 3a, thus confirming that the Fe_7_Co_3_ phase exists in as-deposited nanowires. The generation of other phases in Nd-Fe-B/Fe-Co nanowires must be completed via absorbing the heat annealing process, corresponding to the positive ∆G values. During the multiphase phase transition process, the phase with a smaller energy barrier will preferentially nucleate and grow, then undergo phase transition. According to the thermodynamic calculation results, it can be inferred that the phase nucleation ordering Nd-Fe-B/Fe-Co nanowires during annealing is roughly as follows: a Nd_2_(Fe, Co)_14_B phase with the smallest phase transition energy barrier is preferentially formed, followed by Fe_7_Nd, NdB_6,_ and NdB_4_ phases. It is noted that the ∆G of the Nd_2_(Fe, Co)_14_B phase formed after adding the Co atom is lower than that of the Nd_2_Fe_14_B phase, indicating that the addition of Co reduces the precipitation temperature of the 2:14:1-type phase, thus further promoting the precipitation of more 2:14:1-type hard magnetic phases in composite nanowires, which is consistent with the conclusion proposed by Kojima [30].

The XRD results show that the composite nanowires not only retain the soft magnetic Fe_7_Co_3_ phase formed in an as-deposited state but also stimulate the generation of greater amounts of the hard magnetic Nd_2_(Fe, Co)_14_B phase during the crystallization process. This improves the comprehensive magnetic properties of the nanowires.

### 3.2. Morphology and Microstructure

Figure 5 shows the SEM image of Nd-Fe-B/Fe-Co composite nanowire arrays after partial dissociation from the AAO templates using 5 wt. % NaOH solution. As can be seen from Figure 5a, the Nd-Fe-B/Fe-Co composite nanowires are arranged orderly and uniformly throughout the template range, the deposition efficiency is very high, and the template filling rate is nearly 100%. Figure 5b shows the statistical analysis of the nanowires’ diameter using Nano Measurer software, illustrating that the average diameter of the Nd-Fe-B/Fe-Co nanowire is 71 nm, which corresponds well to the pore diameter of the AAO template. The lateral-view SEM image shown in Figure 5c displays that the length of the nanowires array is approximately 27.01 μm, and the nanowires stand vertically and parallel to each other under the support of the residual AAO templates. The magnified image on the upper right shows the morphology of the nanowires separated from the template holes, which present as elongated and uniformly cylindrical. SEM images present that Nd-Fe-B/Fe-Co nanowires with uniform size are arranged neatly, parallelly, and in a highly ordered fashion. The nanowire energy spectrum analysis results shown in Figure 5d illustrate that the as-deposited Nd-Fe-B/Fe-Co composite nanowires mainly contain Nd, Fe, Co, and B elements, revealing that the metal cations in the electrolyte used for synthesizing the main phases of the nanowire are successfully reduced to atoms and deposited in AAO template pores during the electrochemical deposition process. Figure 6 shows the SEM line scan result of nanowires to further confirm the compositional homogeneity along the nanowire. Figure 6a demonstrates a low-magnification SEM image of Nd-Fe-B/Fe-Co nanowires, and the distribution of the main component elements Fe, Nd, and Co along the nanowires is shown in Figure 6b. The content ratios of Nd, Fe, and Co elements at different positions of 3 μm, 10 μm, and 18 μm are 1:11.23:3.60, 1:9.73:3.01, and 1:10.02:3.28, respectively, illustrating that the relative content of Nd, Fe, and Co is basically consistent along the nanowires. The intensity of every element fluctuates in a tiny range, indicating that the composition distribution of the main elements in nanowires is uniform along the nanowires. 

Figure 7 shows the morphology and lattice structure of deposited Nd-Fe-B/Fe-Co composite nanowires. Figure 7a is the TEM image of the as-deposited nanowires. It can be seen that the dissociated nanowires are uniform elongated cylinders, some of which are agglomerated into bundles, while others are scattered. The illustration in Figure 7a shows that the nanowires possess a uniform diameter of about 70 nm, corresponding with the SEM results. The HRTEM image for zone b of Figure 7a is shown in Figure 7b, and the inserted images in red and blue borders show the Inverse Fast Fourier Transform (IFFT) patterns of regions I and II, respectively. From the IFFT patterns, it can be measured that the d values of regions I and II are 0.2121 nm and 0.2209 nm, both corresponding to the (110) plane of the Fe_7_Co_3_ phase. The measured values are slightly larger than the interplanar spacing of the bcc Fe_7_Co_3_ phase listed in standard X-ray crystallography (d(110)Fe_7_Co_3_ = 0.2025 nm). The larger d values indicate that the RE atom of Nd with a larger radius has existed as a solid solution atom in the Fe_7_Co_3_ phase; therefore, the phase structure of as-deposited Nd-Fe-B/Fe-Co nanowires is the incomplete crystalline Fe_7_Co_3_ phase with Nd atom doping.

Figure 8 displays the TEM analysis of Nd-Fe-B/Fe-Co composite nanowires after annealing at 600 °C for 2 h. Figure 8a is the TEM image for a single nanowire, displaying that the nanowire presents a uniform cylinder-like structure with a smooth outer wall. The selected area electron diffraction (SAED) pattern shown in Figure 8b presents polycrystalline diffraction rings consisting of discrete electron diffraction points, which can be calibrated and analyzed. Based on XRD results, the annealed Nd-Fe-B/Fe Co nanowire is polycrystalline in structure, containing Nd_2_(Fe, Co)_14_B, NdB_4_, NdB_6_, Fe_7_Nd and Fe_7_Co_3_ phases. To confirm the phase composition and distribution of 660 °C-annealed nanowires, zones A–D at different nanowire positions were selected to analyze, and the corresponding results are shown in Figure 9, Figure 10 and Figure 11.

The phase analysis results of zone A, which corresponds to the core portion of nanowires, are shown in Figure 9. It can be seen that the core portion of nanowires is a mixed structure consisting of Nd_2_(Fe, Co)_14_B, Fe_7_Nd phases and a few amorphous phases. The diffraction spots of the FFT patterns for regions I and II shown in Figure 9b,c, respectively correspond to the (20$\overset{\text{¯}}{2}$), (303), (501) and ($\overset{\text{¯}}{2}\overset{\text{¯}}{1}$0), (105) ($\overset{\text{¯}}{1}\overset{\text{¯}}{1}$5) planes of Nd_2_(Fe, Co)_14_B phase with tetragonal structure. The interplanar spacing of the Fe_7_Nd phase with rhombohedral structure is about 0.2971 nm, 0.2835 nm, and 0.2742 nm, corresponding to the (11$\bar{3}$), (104), and (211) planes.

Figure 10a,b show the microstructure of zones B and C, which are located in the outer walls of the nanowire. A two-phase structure consisting of Fe_7_Co_3_ and Nd_2_(Fe, Co)_14_B is observed in Figure 10a. As can be seen from Figure 10b, zone C is composed of NdB_4_ and Nd_2_(Fe, Co)_14_B phases with a straightforward phase interface. The IFFT pattern of phase interface region IV shows many edge dislocations, indicating that the soft magnetic NdB_4_ exhibits a pinning effect relative to the Nd_2_(Fe, Co)_14_B main phase, increasing resistance to domain wall motion.

The FFT patterns for regions I–IV in Figure 11b–e show that zone D exhibits a multiphase structure composed of Nd_2_(Fe, Co)_14_B, NdB_4_, NdB_6_, and Fe_7_Nd phases with a clear phase interface. As shown in the IFFT patterns of regions V and VI, there are a large number of dislocations and dislocation rings at the phase boundary of Nd_2_(Fe, Co)_14_B-NdB_6_ and Nd_2_(Fe, Co)_14_B-NdB_4_, resulting from the pinning effect of NdB_6_ and NdB_4_ to the (primary) main phase. The existence of the pinning effect will hinder the domain wall motion, making it difficult to twist the domain wall in the magnetization process, thereby facilitating and promoting coercivity.

Overall, the annealed Nd-Fe-B/Fe-Co nanowires comprise Nd_2_(Fe, Co)_14_B, Fe_7_Co_3_, NdB_4_, NdB_6,_ and Fe_7_Nd phases, showing a core–shell-like microstructure. Part of the Nd_2_(Fe, Co)_14_B phase and all of the Fe_7_Co_3_ phase are distributed on the outer wall of the nanowire, and the soft–hard magnetic mixture of Nd_2_(Fe, Co)_14_B + NdB_4_ + NdB_6_ + Fe_7_Nd coexists in the core of nanowire.

According to thermodynamic calculation results in Table 2, the ∆G of Nd_2_(Fe, Co)_14_B, NdB_4_, NdB_6_, Fe_7_Nd, and Fe_7_Co_3_ in the nanowires are 29,188.6 J/mol, 90,461.4 J/mol, 84,392.5 J/mol, 82,220.4 J/mol, and −46,742.8 J/mol, respectively. That is, the Fe_7_Co_3_ phase prioritizes spontaneous nucleation and crystallization during electrodeposition. Subsequently, Nd_2_(Fe, Co)_14_B, Fe_7_Nd, NdB_6,_ and NdB_4_ phases are successively formed according to the transition energy barrier for each phase during the heat treatment process.

The phase distribution in nanowires is determined to a great extent by the growth mechanism of the nanowires during electrodeposition (as shown in Figure 12). Prior to electrodeposition, the back of the AAO template is sputtered with a layer of gold film as the conductive electrode. In this process, some gold particles diffuse onto the wall of the template pore, giving it surface absorption energy, promoting the preferential growth of atoms into the tubular structure of the electrode surface and pore wall. Accompanying the formation of the nanotube structure, the surface absorption energy of the nanochannel gradually decreases, making the electric field force the major driving force of electrodeposition. At low current densities, metal atoms pile up from the electrode surface until the nanotubes are completely filled, forming nanowires. Because Fe^2+^ (−0.447 V) and Co^2+^ (−0.28 V) exhibit higher electrode potentials than Nd^3+^ (−2.40 V), Fe^2+^ and Co^2+^ preferentially enter AAO pores under an electric field force and are then reduced on the hole wall, resulting in the following reactions:(4)$${\mathrm{Fe}}^{2+}+{2e}^{-}=\mathrm{Fe}$$
(5)$${\mathrm{Co}}^{2+}+{2e}^{-}=\mathrm{Co}$$

After Fe and Co generate the initial nucleus, the Fe-Co phase is formed on the pore wall; then, Nd^3+^ is induced to conduct co-deposition via the following reaction:(6)$${2\mathrm{Nd}}^{3+}+3\mathrm{Fe}=2\mathrm{Nd}\;(\mathrm{in}\;\mathrm{Nd}-\mathrm{Fe})+{3\mathrm{Fe}}^{2+}$$
(7)$${2\mathrm{Nd}}^{3+}+3\mathrm{Co}=2\mathrm{Nd}\;(\mathrm{in}\;\mathrm{Nd}-\mathrm{Co})+{3\mathrm{Co}}^{2+}$$

Under the combined effect of electric field and surface-absorbed energy, the Fe-Co phase mainly distributes in the outermost layer. Accompanied by heat treatment, a Nd_2_(Fe, Co)_14_B phase with a more minor transition energy barrier first crystallizes at the outer wall, causing the internal components to deviate from 2:14:1. The formation of Fe_7_Co_3_ and Nd_2_(Fe, Co)_14_B phases reduces Fe and Co content in nanowire, providing conditions for the generation of a Nd-rich phase, leading to the crystallization of a Nd-rich phase in nanowire. Meanwhile, the existence of a Nd-rich phase makes the surrounding components close to 2:14:1 again, which results in a mixed structure composed of a hard magnetic Nd_2_(Fe, Co)_14_B phase and soft magnetic NdB_4_ + NdB_6_ + Fe_7_Nd phases inside the nanowires.

### 3.3. Magnetic Properties Analysis 

Figure 13a,b show the hysteresis loops of Fe-Co, Nd-Fe-B, and Nd-Fe-B/Fe-Co composite nanowire arrays before and after annealing at 600 °C for 2 h, respectively. Samples without removing the AAO template were measured in an external magnetic field parallel to the long axis of the nanowires. Figure 13c,d display the dependences of coercivity (Hc) and saturated magnetization (Ms) on nanowire compositions. The magnetic parameters of coercivity (Hc), remanence (Mr), saturated magnetization (Ms), and remanence ratio (Mr/Ms) are listed in Table 3.

Figure 13a displays the hysteresis loops of as-deposited nanowires with different compositions, and the inserted image at the lower right corner is the magnified image of the selected region I. The as-deposited nanowires exhibit prominent soft magnetic properties, with a coercivity of 413.52 Oe for Fe-Co, 366.70 Oe for Nd-Fe-B, and 399.75 Oe for Nd-Fe-B/Fe-Co nanowire arrays. It can be clearly seen from Figure 13c that the Hc for Fe-Co nanowire arrays is slightly higher than that of the other two component nanowires. The Fe-Co nanowires consist of a pure Fe_7_Co_3_ phase, while the phase composition of as-deposited Nd-Fe-B and Nd-Fe-B/Fe-Co nanowires is incompletely crystallized Fe or Fe_7_Co_3_ with a few amorphous and microcrystalline phases, resulting in low crystallinity and slightly lower coercivity.

Figure 13b shows that the magnetic properties of Nd-Fe-B/Fe-Co composite nanowires are significantly improved after annealing, with Hc and Mr significantly increasing and Ms slightly decreasing, showing excellent comprehensive magnetic properties of Hc = 4203.25 Oe, Mr = 68.51 emu/g, Ms = 76.45 emu/g, and Mr/Ms = 0.89. It should be noted that compared with Fe-Co and Nd-Fe-B nanowires, the coercivity of composite nanowires is increased without sacrificing the saturation magnetization, as shown in Figure 13d. The excellent magnetic performances of Nd-Fe-B/Fe-Co nanowires are mainly attributed to the following aspects: (1)The as-deposited Nd-Fe-B/Fe-Co nanowires are soft magnetic Fe_7_Co_3_ and amorphous phases with low Hc. After annealing, the amorphous phase crystallizes into a polycrystalline structure containing Nd_2_(Fe, Co)_14_B, NdB_4_, NdB_6,_ and Fe_7_Nd, while the soft magnetic Fe_7_Co_3_ phase still exists in the nanowires. The generation of the hard magnetic phase Nd_2_(Fe, Co)_14_B with a high anisotropy field hinders the nucleation of the magnetization reversal domain nucleus and the rotation of the magnetic domain, thereby enhancing the coercivity. At the same time, according to the XRD results and Formula (8) [31,32]:(8)$$V_{i}=\left(I_{\max i}/RIR_{i}\rho_{i}\right)/\left(\sum_{i}I_{\max i}/RIR_{i}\rho_{i}\right)\;(i=1,\;2,\;3)$$The content of the hard magnetic Nd2(Fe, Co)14B phase in Nd-Fe-B/Fe-Co composite nanowires prepared via alternating electrodeposition can be calculated as a volume fraction of 0.81, which is significantly higher than that of the hard magnetic phase Nd2Fe14B (volume fraction of 0.73) in Nd-Fe-B nanowires. This is due to the presence of Co ions in the alternating deposition process, which promotes more Nd ions in the electrolyte to be pulled into the AAO template and induces co-deposition, increasing the deposition quantity of Nd atoms and providing favorable conditions for the subsequent generation of a greater amount of hard magnetic phases. In the subsequent heat treatment process, some Co atoms enter the 2:14:1-type main phase to replace the Fe atoms, reducing the crystallization formation energy of the 2:14:1-type hard magnetic phase and contributing to precipitating more hard magnetic phases in nanowires. The coercivity is further enhanced by the increased content of the hard magnetic 2:14:1-type phase.(2)After annealing, the soft and hard magnetic phases intersect with each other in composite nanowires, existing with a large number of edge dislocations at phase interfaces. This causes the domain wall motion in nanowires to be hindered, enhancing the pinning effect between the soft and hard magnetic phases and increasing the pinning resistance, thus increasing the coercivity.(3)The exchange coupling between the hard magnetic Nd_2_(Fe, Co)_14_B phase and soft magnetic phase of Fe_7_Co_3_ will prevent the magnetic moments from orienting along the respective easy magnetization directions and promote the improvement of Mr. According to the exchange spring model proposed by Kneller and Hawig, the exchange-correlation length of the soft magnetic phase can be written as [33]:(9)$$b_{cm}=\pi {\left(A_{s}/2K_{h}\right)}^{1/2}$$wherein *A*_s_ is the exchange energy for the soft magnetic phase, and *K*_h_ is the magnetic crystal anisotropy constant of the hard magnetic phase [34]. In order to achieve sufficiently strong exchange coupling, the grain size for the soft magnetic phase must be less than 2*b_cm_*. According to Formula (8), where *A*(Fe_7_Co_3_) = 1.67 × 10^−11^ J/m [35], K(Nd_2_Fe_14_B) = 4.3 × 10^6^ J/m^3^, it can be calculated that the soft magnetic phase size in the Nd-Fe-B/Fe-Co composite magnet is less than 13.8 nm, which in line with the results observed in the TEM image. Therefore, there is a strong exchange coupling effect in soft–hard magnetic phases in annealed nanowires, facilitating to increase Ms. Based on the Stoner Wohlfarth theory, the phenomenon of remanence enhancement will occur in the magnet under the coupling of soft and hard magnetic exchange when Mr/Ms is much higher than 0.5. As the data listed in Table 3, the Mr/Ms of annealed Nd-Fe-B/Fe-Co composite nanowires is as high as 0.89, therefore exhibiting the magnetic characteristics of high remanence and a high magnetic energy product of 28 MGOe under strong exchange coupling.

It can be seen from the magnetic properties analysis that the characteristics of Nd-Fe-B/Fe-Co magnetic nanowires are mainly controlled via the magnetic exchange coupling mechanism with the assistance of the nucleation field and pinning field. The presence of the Fe_7_Co_3_ phase and the strong exchange coupling of soft and hard magnetic phases increase Hc and produce a remanent magnetic enhancement effect, showing a demagnetization curve similar to that of the hard magnetic phase, and the magnetic energy product is significantly increased. The magnetic performance of Nd-Fe-B magnetic nanowires has been significantly improved in comparison with those obtained in previous research. That is to say, compositing with a soft magnetic Fe-Co alloy is an effectual route to enhance the comprehensive magnetic properties of Nd-Fe-B-based nanowires, rendering it more suitable for application in magnetic storage media.

## 4. Conclusions

In conclusion, highly ordered Nd-Fe-B/Fe-Co hybrid magnetic nanowires with cylinder-like structures were fabricated in porous AAO templates by means of alternating electrochemical deposition in Nd-Fe-B and Fe-Co aqueous solution. The as-deposited Nd-Fe-B/Fe-Co hybrid nanowires are composed of Nd-atom-doped incompletely crystallized Fe_7_Co_3_ and amorphous phases, which have typical soft magnetic properties. The annealing treatment recombines the atoms to form a polycrystalline structure containing hard magnetic Nd_2_(Fe, Co)_14_B and soft magnetic NdB_4_, NdB_6_, Fe_7_Nd, and Fe_7_Co_3_ phases. The nucleation sequence of each phase in nanowires is as follows: the Fe_7_Co_3_ phase prioritizes spontaneous nucleation and crystallization in as-deposited nanowires. Subsequently, the Nd_2_(Fe, Co)_14_B phase preferentially nucleates, followed by Fe_7_Nd + NdB_6_ + NdB_4,_ in order, during the heat treatment process. The structural characterization revealed that part of the Nd_2_(Fe, Co)_14_B and all of the Fe_7_Co_3_ phase distribute in the outer wall of a nanowire; the soft–hard magnetic mixture of Nd_2_(Fe, Co)_14_B + NdB_4_ + NdB_6_ + Fe_7_Nd coexists in the core of nanowire. The coercivity improvement after annealing is mainly due to the formation of a hard magnetic Nd_2_(Fe, Co)_14_B phase with a high anisotropy field, hindering the nucleation of the magnetization reversal domain nucleus and the rotation of the domain wall, thereby heightening the nucleation field of the annealed nanowires.

Compared with Nd-Fe-B nanowires, the comprehensive magnetic properties of Nd-Fe-B/Fe-Co composite nanowires are significantly improved with a coercivity of 4203.25 Oe and a saturation magnetization of 76.45 emu/g. The excellent magnetic performances can be attributed to the generation of a greater amount of the hard magnetic Nd_2_(Fe, Co)_14_B phase in composite nanowires, the pinning effect between different phases, and the remanence enhancement effect generated by the strong exchange coupling of hard and soft magnetic phase after being combined with Fe-Co alloy, cooperatively resulting in a remanence ratio as high as 0.89 and a high magnetic energy product of 28 MGOe. 

Therefore, synthesizing Nd-Fe-B/Fe-Co composite nanowires through alternating electrochemical deposition is an effectual method to improve coercivity without sacrificing remanence. And the RE-Fe-B-based nanowires have a broad application prospect in the field of magnetic memory materials.

## Figures and Tables

**Figure 1 materials-16-05541-f001:**
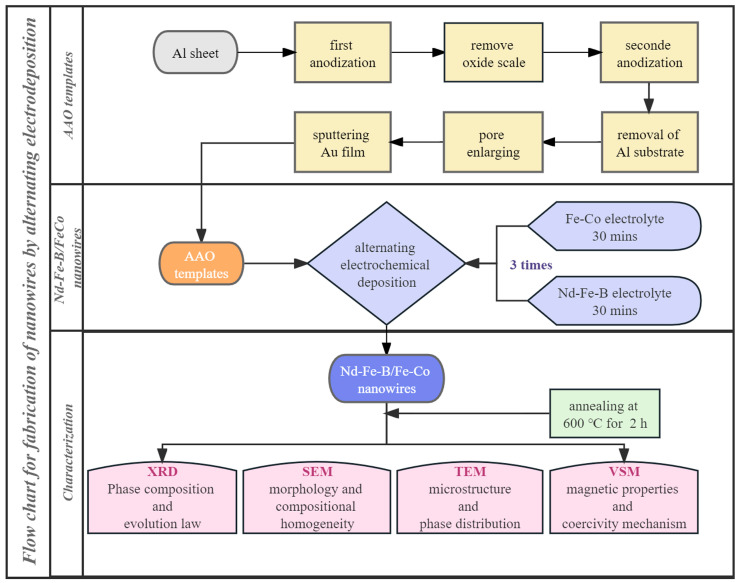
The flow chart describing the procedure for preparing nanowires based on electrochemical deposition in an AAO template.

**Figure 2 materials-16-05541-f002:**
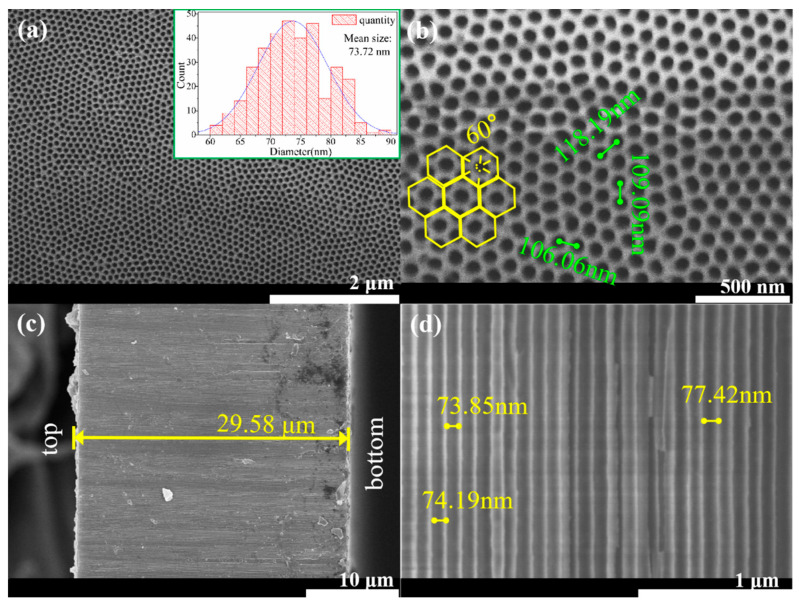
SEM images of AAO template prepared via two-step oxidation method: (**a**) top-view SEM image, the inset showing pore size distribution in AAO template; (**b**) top-view SEM image in high magnification; (**c**) cross-sectional SEM image; (**d**) cross-sectional SEM image in high magnification.

**Figure 3 materials-16-05541-f003:**
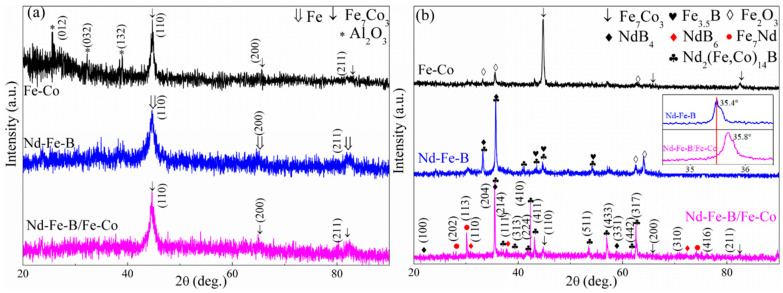
XRD patterns of nanowires with different compositions: (**a**) as-deposited; (**b**) annealed at 600 °C for 2 h. The inserted image is the magnification view of the diffraction peak for the (204) plane at the angle of 34.5°–36.5°.

**Figure 4 materials-16-05541-f004:**
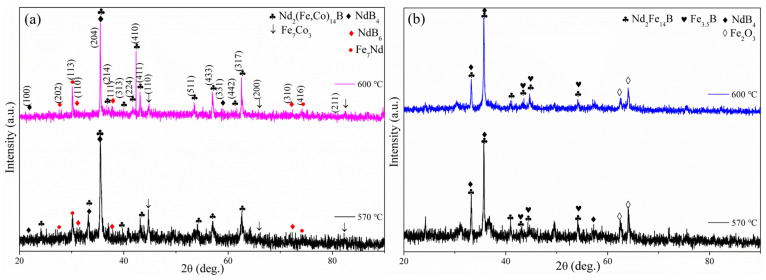
XRD patterns of nanowires with different compositions after annealing at 600 °C and 570 °C: (**a**) Nd-Fe-B/Fe-Co nanowires; (**b**) Nd-Fe-B nanowires.

**Figure 5 materials-16-05541-f005:**
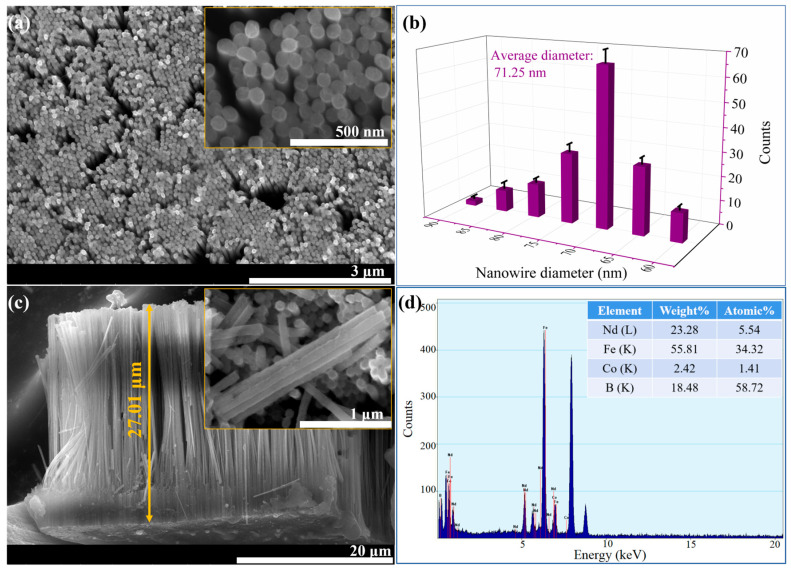
SEM images of Nd-Fe-B/Fe-Co nanowires prepared by means of the alternate electrodeposition method: (**a**) top-view SEM image—the illustration in the upper right corner is the high magnification SEM image; (**b**) the diameter statistical distribution diagram of nanowires; (**c**) cross-sectional SEM image; (**d**) EDS spectrum of nanowires.

**Figure 6 materials-16-05541-f006:**
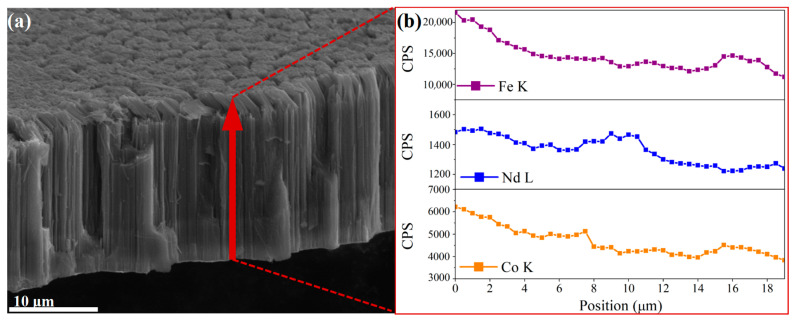
(**a**) SEM image of as-deposited Nd-Fe-B/Fe-Co nanowires; (**b**) the SEM-EDS line scan along the long axis of nanowire (red line in the SEM image).

**Figure 7 materials-16-05541-f007:**
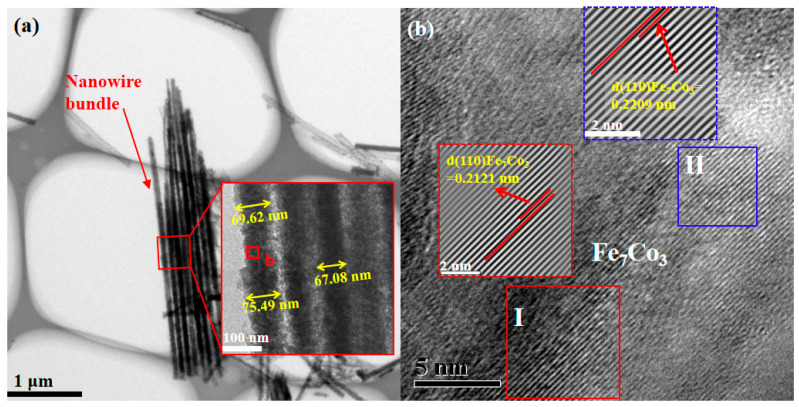
TEM observation and analysis of as-deposited Nd-Fe-B/Fe-Co nanowires: (**a**) TEM image of nanowire bundle, and the inserted image is the magnified view of the selected portion with a red border; (**b**) HRTEM image of zone b in (**a**) and the corresponding IFFT patterns of regions I and II.

**Figure 8 materials-16-05541-f008:**
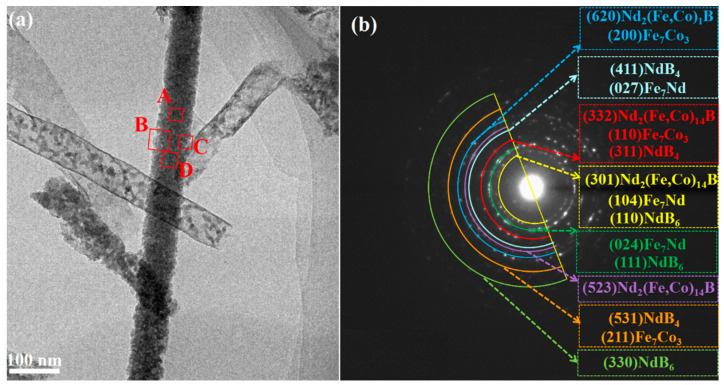
TEM observation and analysis of Nd-Fe-B/Fe-Co nanowires after annealing at 600 °C for 2 h: (**a**) the morphology of a single composite nanowire; (**b**) SAED pattern of the annealed nanowire.

**Figure 9 materials-16-05541-f009:**
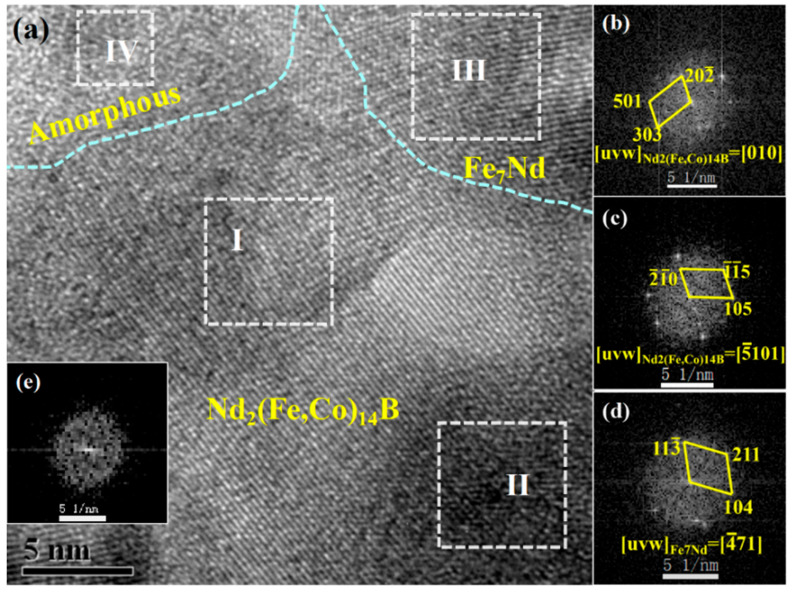
HRTEM analysis of annealed Nd-Fe-B/Fe-Co nanowires: (**a**) HRTEM of zone A in Figure 8a; (**b**–**e**) the corresponding FFT patterns of regions I–IV.

**Figure 10 materials-16-05541-f010:**
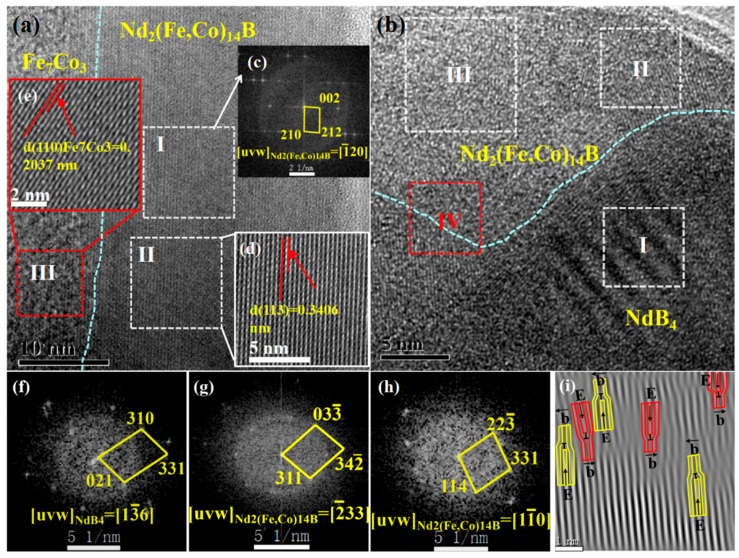
HRTEM analysis of annealed Nd-Fe-B/Fe-Co nanowires: (**a**,**b**) HRTEM images of zones B–C in Figure 8a; (**c**) FFT pattern of the selected region I in (**a**); (**d**,**e**) the magnified view of regions II–III in (**a**); (**f**–**h**) FFT patterns of regions I–III in (**b**); (**i**) the corresponding IFFT pattern of region IV in (**b**).

**Figure 11 materials-16-05541-f011:**
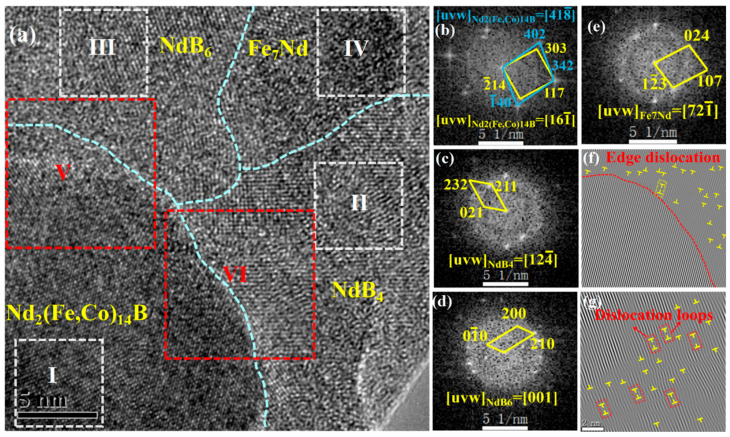
HRTEM analysis of annealed Nd-Fe-B/Fe-Co nanowires: (**a**) HRTEM image for zone D in Figure 8a; (**b**–**e**) FFT patterns of the selected regions I–IV in (**a**); (**f**,**g**) the corresponding IFFT patterns of regions V–VI in (**a**).

**Figure 12 materials-16-05541-f012:**
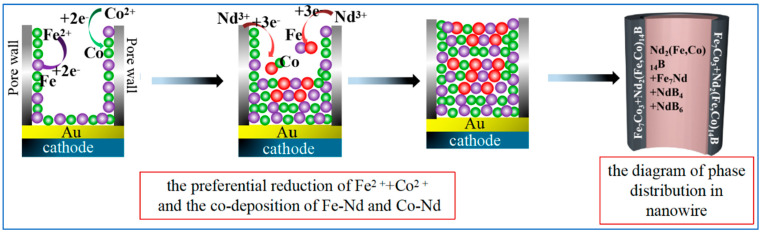
Schematic illustrations for the electrodeposition growth mechanism of Nd-Fe-B/Fe-Co composite nanowires in AAO pores and the phase distribution in nanowires.

**Figure 13 materials-16-05541-f013:**
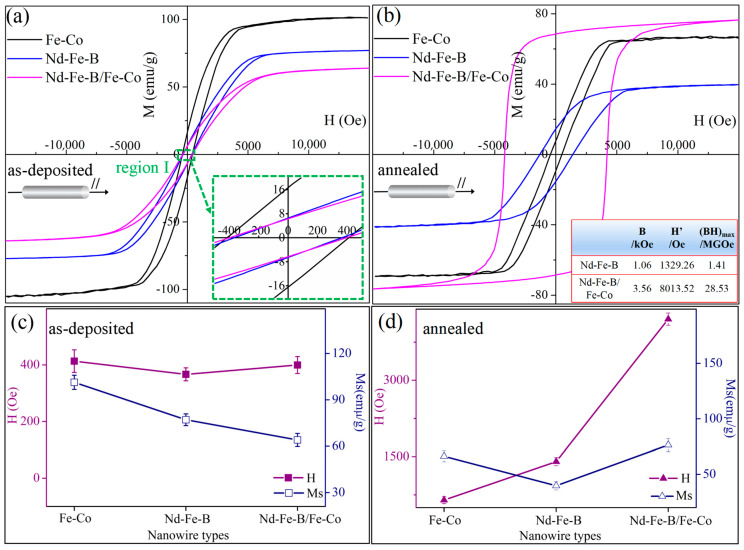
Magnetic hysteresis loops of nanowires with different compositions measured at room temperature under external magnetic field applied direction parallel to the long axis of nanowires: (**a**) as-deposited; (**b**) annealed at 600 °C for 2 h; (**c**) the variation trends of Hc and Ms for as-deposited nanowires with different compositions; (**d**) the variation trends of Hc and Ms for 660 °C-annealed nanowires with different compositions.

**Table 1 materials-16-05541-t001:** The electrolyte composition of Nd-Fe-B/Fe-Co composite nanowires prepared via alternate deposition.

Electrolyte	Concentration/(mol/L)							
	NdCl_3_	FeCl_2_	CoCl_2_	C_6_H_8_O_7_	C_6_H_8_O_6_	H_3_BO_3_	C_2_H_5_NO_2_	NH_4_Cl
Fe-Co	0	0.35	0.15	0.073	0.007	0.485	0	0
Nd-Fe-B	0.03	0.2	0	0	0.007	0.485	0.4	0.5

**Table 2 materials-16-05541-t002:** The maximum peak intensities of each phase in Nd-Fe-B/Fe-Co and Nd-Fe-B nanowires after annealing at 600 °C and 570 °C, and the corresponding Gibbs free energy.

Specimen	Phase	*I* _max_		∆*G* = ∆*H*/(J/mol)
		600 °C	570 °C	
Nd-Fe-B/Fe-Co	Nd_2_(Fe, Co)_14_B	347	300	29,188.6
	Fe_7_Nd	138	91	82,220.4
	Fe_7_Co_3_	83	104	−46,742.8
	NdB_4_	68	44	90,461.4
	NdB_6_	78	53	84,392.5
Nd-Fe-B	Nd_2_Fe_14_B	271	230	33,264.0
	NdB_4_	108	58	125,723.9
	Fe_3_._5_B	72	59	43,845.8

**Table 3 materials-16-05541-t003:** The magnetic parameters of nanowires with different compositions.

Sample	State	H_c_/Oe	M_r_/(emu/g)	M_s_/(emu/g)	M_r_/M_s_
Fe-Co	as-deposited	413.52	15.26	101.29	0.15
	annealed	648.56	10.28	66.28	0.16
Nd-Fe-B	as-deposited	366.70	6.55	77.10	0.09
	annealed	1404.32	15.97	39.74	0.40
Nd-Fe-B/Fe-Co	as-deposited	399.75	6.31	64.01	0.10
	annealed	4203.25	68.51	76.45	0.89

## Data Availability

Any further detailed data may be obtained from the authors upon reasonable request.

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
