# Peer review of "Study on the Microstructure and Magnetic Properties of Nd-Fe-B/Fe-Co Composite Nanowires"

_materials, 2023, doi:10.3390/ma16165541_

Round 1

Reviewer 1 Report

The current work illustrates the microstructure and magnetic properties of Nd-Fe-B/Fe-Co composite nanowires. The presentation of the current data is very good. In addition, a deep microstructure analysis is well-studied and presented. I found the manuscript very interesting and will make a good contribution to the scientific community. The high value of Hc and Mr obtained in the current works makes it very promising for many applications.  Thus, I recommend accepting the manuscript to be published in Materials, after the authors maintain these necessary points.

The introduction.

The introduction strongly needs to be improved with more information about soft magnetic nanowires and their applications in various fields, especially spintronic applications. In addition, it will be great if they spot line the gap in this research point and the importance of the current study to fill this gap.

 Materials and methods:

1-      The authors should add more information and details in the experimental part to show the preparations of AAO templates and the steps of fabrication of Nd-Fe-B/Fe-Co nanowires.

2-      The authors should add more details about the selected annealing condition of the current sample. What about lower /higher temperature & time annealing? Please describe this point in detail in the revised version of the manuscript.

 Results:

1-          The authors should add the uncertainties for all outcome data in the tables and the figures to see the real behavior of all parameters studied in the current manuscript.

2-          It will be great if the authors add the distributions of the elements at least for the as-deposited sample to confirm the homogeneity of the sample.

3-          Kindly add the references for all equations presented in the manuscript.

4-          The references need to be enhanced with more recent works with other research groups.

Reviewer 2 Report

The paper “Study on the microstructure and magnetic properties of Nd-Fe-B/Fe-Co composite nanowires” is devoted to preparation and investigation of arrays of Nd-Fe-B/Fe-Co magnetic nanowires obtained by alternate deposition from a Fe-Co solution. XRD, FESEM, TEM, EDS, VSM techniques were used for the samples characterization.  The work is of scientific interest to specialists in the field of obtaining magnetic composite materials for functional applications. The data are reliable and do not cause much doubt. Nevertheless, there are several points before the paper can be published. I hope that authors after major revisions can improve the paper and can publish it in Materials.

  1. Introduction part should be improved by new relevant literature in the field of magnetic nanowires fabrication. I suggest to use the following references (see and discuss:

https://doi.org/10.1134/S1063739719020100;

https://doi.org/10.1016/j.jmmm.2019.03.016;

https://doi.org/10.3390/nano11071775.

  1. Don’t leave large gaps between formulas and text.
  2. Tables and figures should be aligned at the same level as the text (to facilitate the copy-editing of larger tables, smaller fonts may be used, but no less than 8 pt. in size).
  3. Please explain why did you use a voltage of 40 V for obtaining of ААО templates?
  4. For a complete understanding of the production of composite nanowires, describe the experimental part in more detail.
  5. More practical recommendations should be added in the manuscript.
  6. All ion charges must be in superscript.
  7. All text must be under the same alignment.
  8. Conclusion should be re-written more widely.

Minor editing of English language required

Reviewer 3 Report

Comments

1.       The abstract needed to be restructured, it was too obstruct. The main idea of the research cannot be understood, and the problem statement and the general introduction are not spelled out.

2.       State the main findings of the research and the novelty of this research in the last paragraph of the abstract.

3.       You made mentioned “microstructure; electrochemical deposition” in the keywords but I cannot see these two words in your abstract, you must follow the scientific format of paper writing.

4.       The introduction part is not constructive, kindly add more information from different literature.

5.       Include the methodology flow chart in section 2.1.

6.       There is too much noise in figure 1, remove the noise and give a more detailed explanation, you can see the suggested paper https://doi.org/10.1016/j.jmrt.2020.11.053

7.       The authors should use a maximum of 11 words per sentence. In this way, the authors can be clear and concise.

8.       Kindly check and correct grammatical mistakes

9.       Re-write the conclusion part in a better clarity form in a concise illustration.

Extensive editing of English language required

Round 2

Reviewer 1 Report

The authors fixed most of my comments and the manuscript can be accepted in the present form.

Author Response

Thank you for your approval!

Reviewer 2 Report

Now the paper can be accepted in its present form. 

Author Response

Thank you for your approval!

Reviewer 3 Report

1.       I discovered that there are sections that seem to be unoriginal, having appeared in previously published work(s). In this case, the overlap goes beyond the normal occurrence of standard phrases in your field.

2.       In section 2.1 of the manuscript the authors claim that they have responded to the question of using flowcharts to explain the details of the experimental procedure. It seems the authors did not know the difference between a flowchart and a schematic diagram.

3.       Main issues are not addressed in the manuscript which still make the manuscript in bad shape.

4.       The Turnitin is 33%, and the originality of the manuscript is questioned.

5.       Poor grammatically used and inconsistency. 

 Extensive editing of the English language required

Round 3

Reviewer 3 Report

Nil

 Minor editing of the English language required